# Gender diverse people's psychological wellbeing and identity in the context of gender affirming speech pathology practice: A qualitative study protocol

**Julia Tanase**[1,2]*, **Sterling Quinn**[1], **Sabine Corsten**[3], **Hilke Hansen**[2], **Jennifer Oates**[1]

**1** School of Allied Health, Human Services & Sport, La Trobe University, Bundoora, Australia, **2** Faculty of Business Management and Social Sciences, University of Applied Sciences Osnabrück, Osnabrück, Germany, **3** Department of Healthcare and Nursing, Catholic University of Applied Sciences Mainz, Mainz, Germany

* j.tanase@latrobe.edu.au

**Data Availability Statement:** No datasets were generated or analysed during the current study. All

## Abstract

### Background

Gender diverse people may experience discomfort with the sound of their voice. Additionally, their psychological wellbeing, closely connected to a person's identity, or perspective of themselves, is often reduced. A primary goal of gender affirming healthcare is to support clients' psychological wellbeing. Speech pathology practices assist clients to alter their voice to develop authentic self-presentation. These practices have been shown to have positive outcomes related to clients' voices but have not yet been shown to improve overall client psychological wellbeing. Assisting gender diverse people to transform a negative identity perspective into a positive one could have a beneficial impact on their psychological wellbeing. Therefore, to support clients' psychological wellbeing, gender affirming speech pathology care could benefit from focusing on gender diverse clients' identity more broadly, i.e., beyond gender. For this type of care, speech pathologists would need to see themselves as responsible for supporting client psychological wellbeing and identity. However, some clinicians may be hesitant to incorporate more holistic approaches to provide such care. This study aims to explore gender diverse speech pathology clients' views on psychological wellbeing and identity and speech pathologists' perspectives and actions in supporting clients in that regard.

### Methods

This qualitative study approaches the research topic through insights from gender diverse people's and speech pathologists' subjective perspectives and experiences on psychological wellbeing and identity. Gender diverse people will participate in one-on-one episodic interviews, whereas speech pathologists providing gender affirming care will participate in focus group discussions. Data will be analysed with reflexive thematic analysis. The study is supported by collaborators from the gender diverse community. Study findings will be

relevant data from this study will be made available upon study completion.

**Funding:** This work is supported by a La Trobe University Graduate Research Scholarship and a La Trobe University Full Fee Research Scholarship (https://www.latrobe.edu.au) that the first author (JT) holds. The funder had no role in study design, data collection and analysis, decision to publish, or preparation of the manuscript.

**Competing interests:** The authors have declared that no competing interests exist.

disseminated in an accessible manner to healthcare professionals providing gender affirming care, mainly speech pathologists, and to the gender diverse community.

## Implications

Study findings are anticipated to contribute to further understanding gender diverse people's psychological wellbeing and identity in a gender affirming speech pathology context to tailor practices to the unique needs of gender diverse clients.

## 1 Background

People who identify as gender diverse, an umbrella term used to refer to a range of gender identities (e.g., transgender, non-binary, gender queer), experience their gender identity differently to their presumed gender category at birth [1]. In addition, they may wish to present their gender differently to this category [1]. Some gender diverse people do not feel they can present their gender authentically through their vocal patterns, which can lead to feelings of discomfort or distress [2]. Vocal patterns refer to the sound of a person's voice (vocal pitch, intonation, oral resonance, loudness, vocal quality) and their speaking prosody (tempo, stress) [3, 4]. Moreover, gender diverse people's psychological wellbeing (PWB) is often reduced, and they show higher rates of depression, anxiety, substance abuse, suicidal thoughts, and suicide attempts than cisgender people who do identify with their gender presumed at birth [5–10].

A person experiences positive PWB if they subjectively evaluate their life as overall positive [11] and if the person's resources and challenges in life are in balance [12]. PWB can be conceptualised from a hedonic or eudaimonic perspective [11]. The hedonic conceptualisation distinguishes between satisfaction in life, positive affect and negative affect [11]. The eudaimonic conceptualisation of PWB distinguishes between six different dimensions, i.e., experiencing purpose in life, the ability to adjust one's environment according to one's own needs, positive relationships with others, autonomy, personal growth, and an overall positive perspective on oneself [11, 13]. The eudaimonic perspective takes into account that people may have to negotiate challenges throughout their lifespan in order to give meaning to such events [11, 14]. This meaning-making process is closely connected to how a person sees themselves and to a person's PWB [14]. Therefore, the eudaimonic conceptualisation of PWB seems appropriate to understand gender diverse people's PWB. The following paragraphs elaborate on both gender diverse people's potential challenges and their perspective on themselves.

One explanation for gender diverse people's reduced PWB is the challenge of gender minority stress they face [15]. Due to their marginalised–or minority–position, they may experience external stressors such as misgendering, physical and verbal harassment or being refused healthcare treatment [10, 16]. This may lead to gender diverse people not disclosing their gender diverse identity to others to avoid these external stressors [15]. As a result of gender minority stressors, gender diverse people may develop internalised transphobia, a discomfort with and negative view of their gender identity [15, 17]. Both, nondisclosure of identity and internalised transphobia further contribute to negative PWB [15, 17, 18]. At the same time, gender diverse people can develop resources to mitigate the effects of gender minority stress on their PWB. This can include having a supportive environment and feeling connected to others [15]. Feeling authentic and openly presenting one's experienced gender identity to others can act as a further resource for PWB [19]. Additionally, experiencing identity pride, a feeling of self-worth and positive view of one's identity further contributes to PWB [15, 20].

As elaborated in the gender minority stress theory, gender diverse people's PWB is closely linked to their view of their own identity [15]. Identity can be understood as a person's subjective, constructed and evolving story of the self, or their narrative identity [21, 22]. A person constructs their identity through an ongoing interplay of their own meaning-making of life events, their internal stories about themselves, some of which they share with others, as well as through stories about themselves told by others. The later includes broader stories or narratives about how to live a life, that are culturally shared within a broader community [21–23]. This ongoing, individual and interindividual meaning-making means that a person's identity is in constant change and therefore mutable [24]. Since gender diverse people are often stigmatised in Western societies [1, 25], some culturally shared stories about them are negative. Gender diverse people may integrate the values, beliefs, and expectations these stories contain into their own value system [17, 26]. Looking at identity as a person's story and knowing that gender diverse people are often marginalised, makes it evident that it can be particularly easy for gender diverse people to develop a negative story of their self.

Supporting gender diverse people's PWB is increasingly seen as a key goal of gender affirming healthcare [1]. This is also the case for speech pathology (SP; hereafter used to refer to the profession) practices, both from speech pathologists' [1, 27–29] and gender diverse clients' perspectives [2]. If gender diverse people feel they cannot present their gender authentically through their vocal patterns, they may experience discomfort or distress and seek professional support from a speech pathologist [2]. This can be the case for any gender diverse person, e.g., transfeminine, transmasculine, or non-binary [1], even for people whose vocal patterns may have changed through gender affirming hormones [30, 31]. Gender affirming SP practices extensively reported to date aim to assist the gender diverse client to develop authentic self-presentation [4]. Authentic self-presentation can mean that while some gender diverse people want to present themselves so that others see them as cisgender female or cisgender male, some do not want to present in this way [32]. This shows that client needs are highly individual and reflects the heterogeneity of the gender diverse community. To achieve authentic self-presentation, speech pathologists help clients to modify their vocal patterns and to transfer their new skills into everyday life [4]. This approach is in line with currently recommended and described gender affirming SP practices from Western countries such as Germany, Sweden, and Australia, which convey a somewhat narrow view on gender affirming care, prioritising gender presentation through vocal patterns [29, 33]. These practices of addressing vocal patterns to develop authentic self-presentation also meet an often expressed need of gender diverse SP clients [4, 29].

Gender affirming SP practices have been shown to have positive outcomes for the client, such as increases in client satisfaction with their vocal patterns [29, 34–40], confidence in using newly learned vocal skills [34], self-confidence [39], and social participation [35]. However, an intense engagement with their own vocal patterns may result in gender diverse clients experiencing distress, which can limit their capacity to engage in SP, even though they wish to do so [2]. While current research shows positive effects of gender affirming SP practices, these outcomes relate to clients' vocal patterns only and have not yet been shown to lead to an overall improvement in client PWB [35]. Therefore and despite the positive outcomes of gender affirming SP practices, addressing client PWB is not only important to support the client in achieving their SP-specific goals, but also to meet the overall goal of enhancing PWB through gender affirming healthcare.

Additional to professional training in gender affirming voice modification approaches, a full understanding of a gender diverse client's individual challenges and resources in life is crucial when providing gender affirming care that aims to support client PWB [1]. Current suggestions to address client PWB in gender affirming SP care are to include counselling [29, 40,

41] and assist the client to create resources for managing distress [28]. This could involve creating a supportive, accepting SP setting [2], developing responses to instances of misgendering [42], or reflecting on gender-norms [2]. Another suggestion is to discuss the client's internalised values, beliefs, and priorities as related to their current and desired vocal patterns [43, 44], or what gender means for the client in relation to those patterns [27]. These suggested practices go beyond working on bodily functions such as vocal patterns and are inclusive of the broad range of gender identities. However, these emerging practices mainly focus on centring vocal patterns and gender as one aspect of gender diverse people's identity. Even though suggested approaches progressed from working on bodily functions to aspects of identity, approaches that centre a person's whole identity could be a way to further support client PWB.

Research in psychology is beginning to show that creating positive narratives around gender diverse identities can have positive effects on gender diverse people's PWB [19, 45]. As a person can re-author negative stories that shape their view of themselves into more positive ones [46], it can be hypothesised that gender affirming SP care could benefit from practices that focus less on gender presentation or vocal patterns than current approaches do. For example, when considering the different phases of gender diverse people's identity development, a crucial point for a more positive story could be reached when identifying as gender diverse becomes just one part of a person's whole identity story [17]. Gender could then be integrated into a larger identity story, which makes gender labels less important and potential existing ambiguity around a person's gender identity more tolerable [17]. Gender affirming SP care could centre what experiences, values, and beliefs shape gender diverse clients' identity in a broader sense. This may be a way to assist clients to transform a negative view on their identity into a positive one, which could have a positive effect on their PWB. To date, there is a lack of research on PWB, identity and SP, particularly research that explores these topics from the gender diverse clients' subjective view.

The type of care clinical professionals provide is connected to how they see themselves in a professional context [47]. This perspective is influenced by clinicians' skills, knowledge, their (professional) socialisation, and the context in which they practice their profession [47]. For instance, speech pathologists who work in interdisciplinary teams at gender affirming healthcare clinics may be more aware of PWB as an overall goal of gender affirming healthcare as opposed to clinicians working in single discipline settings with peer speech pathologists. On the other hand, speech pathologists working in gender affirming interdisciplinary teams may feel less responsible for client PWB as they know that mental health professionals on the team are likely to work on client PWB. Currently, it is unclear how the professional setting influences speech pathologists' mindset regarding the support of client PWB and identity.

Further, as DiLollo and Neimeyer [23] suggest, speech pathologists seem to be hesitant to incorporate more holistic approaches like counselling that go beyond the training of client skills, even though clinical guidelines suggest otherwise [23]. Their suggestion is supported by SP research in the field of aphasia rehabilitation that shows that there seems to be some uncertainty about speech pathologists' scope of practice and some clinicians may even feel conflicted when counselling clients with aphasia [48]. Literature that explores speech pathologists' perspectives on themselves and their work in gender affirming care has been sparse [see 49–51 for example]. To date, no study has been reported that centres speech pathologists' perspectives on their own role as clinicians and their actions in supporting client PWB and identity in a gender affirming context.

Despite new research trends and suggestions for gender affirming clinical practices in SP, there has been limited research exploring PWB and identity in this context of SP practices. Therefore, the following research questions will be investigated:

1. What enables gender diverse people, who seek support from a speech pathologist, to experience their identity as positive?

2. What do gender diverse people, who seek support from a speech pathologist, experience as (not) valuable in a SP setting?

3. How do speech pathologists see their role in supporting gender diverse clients' PWB and experience of identity?

4. What do speech pathologists report to do to support gender diverse clients' PWB and experience of identity?

Since this is a qualitative project, the research process is dynamic and some aspects of the project may change during the course of the research [52]. For example, the research questions outlined above could be further refined, or the data collection methods may be adapted. Comparing this study protocol with later publications arising from the study will demonstrate this dynamic process.

This protocol paper is directed towards researchers who would like to follow a reflexive approach in their qualitative research. Following good quality practice [53], the study protocol provides detailed description of the research approach and can serve as methodological orientation for researchers in the area [54]. Conceptualising identity as flexible and PWB as multifaceted as outlined here can be a useful starting point for researchers to design future studies that deepen understanding of identity and PWB for gender affirming healthcare services.

## 2 Materials and methods

Ethical approval has been obtained for this study from La Trobe Human Research Ethics Committee on 13 March 2023 (approval number HEC23004). The study protocol describes the different steps in this study, following the Reflexive Thematic Analysis Reporting Guidelines (RTARG [53]; see S1 Appendix for the RTARG filled in for this study).

### 2.1 Reflexivity statement

This research is shaped by the diverse professional and personal background of the members of the research team, their current knowledge, and ongoing reflections about the study.

The first author (JT) is a middle-class academic who has been predominantly trained in quantitative research approaches and is relatively new to qualitative research. She holds a master's degree in SP and has conducted research in gender affirming care in the past. JT has worked clinically in gender affirming care in SP private practices in Germany and has been trained more recently in narrative practices. She previously lived in Germany, is a native German speaker and identifies as queer and cisgender. This positionality makes JT an ally and outsider researcher for the gender diverse community, but an insider researcher for the community of speech pathologists providing gender affirming care [55].

The remaining authors on the research team, SQ, SC, HH, and JO, are a group of four middle-class academics, each holding a PhD and being trained in speech pathology. One person has an additional background in sociology, one in psychology and one in identity-focussed research in SP. Two of these authors have worked as researchers and clinicians in gender affirming care in Australia (both within and outside of interdisciplinary healthcare settings), two are native English speakers and two are native German speakers. One of these authors identifies as gender diverse.

From a theoretical perspective, this study is shaped by research and practices across multiple disciplines. Key influential elements are the gender minority stress and resilience theory

[15, 56, 57], theories about PWB [13, 58], narrative identity and identity development [17, 21, 22, 59], counselling approaches and narrative practice [23, 46, 60], clinicians' perspectives on their professional role [47], as well as emerging discussions around PWB and holistic approaches in gender affirming SP services [2, 27, 28, 42]. Linking these theories as outlined in the background section of this protocol have influenced the design of this study.

## 2.2 Setting of the study

This study is part of a PhD candidature in SP at La Trobe University, Melbourne/Australia with JT as a doctoral candidate and JO, SQ and HH as her supervisors.

JT (first author) will be the person on the research team interacting with all study participants and study collaborators throughout the study. Participants and collaborators will be all German-speaking and located in a German speaking country. Since the first author formerly lived and worked in Germany, she shares a cultural and linguistic background with the study participants and collaborators. This will support her to immerse herself into the participants' and collaborators' cultural and linguistic context and to react flexibly to them. This is considered particularly important because of sensitive topics such as identity and PWB that will be discussed.

Since the first author is now based in Australia, all interactions will take place online. The General Data Protection Regulation (GDPR) of the European Union [61] has been considered in selecting and working with all online platforms and software (see S2 Appendix).

## 2.3 Methodology and study design

As outlined above, there is a lack of in-depth, person-centred research around identity and PWB in gender affirming SP clinical care. Here, 'clinical' refers to any healthcare setting providing direct client support, i.e., private practices, or healthcare clinics. This qualitative study is designed to centre experiences and meaning-making [62, 63] of gender diverse people and speech pathologists about the research topic.

The study design is based on a constructivist paradigm with the belief that there are multiple realities, interpreted by each person individually and that knowledge is commonly constructed by the researcher and research participant [64, 65]. Episodic interviews [66] will be conducted with gender diverse people to gain a detailed insight into their personal experiences on the research topic. Focus group discussions [67] with speech pathologists will be conducted to facilitate intense discussion about the research topic. Data will be collected in two phases: After the first three interviews and after the first focus group discussion, data collection will be paused so that there will be time to reflect on the data collection process and decide whether there is a need to adjust it before any further data collection. Data will be analysed using reflexive thematic analysis (RTA) [68].

To further counteract deficit-oriented discourses in research with gender diverse people [69], the study is designed in a way that acknowledges both the challenges and the resources associated with identifying as gender diverse. For instance, the study goals are not directed towards gender diverse people's negative experiences only, as is reflected in the open choice of wording during data collection and the planned non-deficit-oriented way of reporting on the data.

Additionally, a collaborative research approach [70] is applied, aiming to conduct research that is likely to be perceived as meaningful and beneficial by the gender diverse community and for speech pathologists providing gender affirming care. Therefore, members of the gender diverse community are actively involved as collaborators.

## 2.4 Study participants and collaborators

Gender diverse people and speech pathologists will be invited as study participants to share their perspectives and experiences on the research topic, either in one-on-one interviews or in focus groups discussions. Additionally, gender diverse people who form an Advisory Group (AG; hereafter used interchangeably with 'collaborators') will collaborate on the project to provide support throughout.

**2.4.1 Inclusion and exclusion criteria.** A comprehensive list of inclusion and exclusion criteria is given in Table 1. The following section elaborates only on the inclusion criteria that need further explanation as to their rationale. People known to the research team will be excluded to avoid the risk that people's personal relationship with members of the research team coerces them to participate in this study and to avoid evoking socially desirable responses from study participants. All study participants and collaborators need to be fluent in German since any interaction with them will take place in German.

*Gender diverse people.* Gender diverse people who want to participate should have attended at least five gender affirming SP sessions. Since gender diverse study participants may be challenging to recruit, the minimum number of SP sessions to participate in this study is half of the number of SP sessions commonly prescribed for gender affirming care in Germany [71, 72]. In general, the literature does not specify any particular number of SP sessions for gender diverse clients. However, for clients who aim to present as female, a recent review reports a minimum number of five SP sessions [73]. It can be assumed that this minimum number is transferable to any gender diverse person who seeks professional support from a speech pathologist, regardless of whether their vocal patterns have changed through gender affirming hormones. Hence, the inclusion criterion of a minimum of five SP sessions will apply to any gender diverse person interested in participating in the study.

*Speech pathologists.* Due to the reported low numbers of speech pathologists who have received training or have been working in gender affirming care [50, 51], no minimum amount of experience will be required for speech pathologists to participate in this study. Speech pathologists who are likely to be considered experts by peer speech pathologists working in gender affirming care in Germany will be excluded. This encompasses speech

**Table 1. Inclusion and exclusion criteria for the study participants and collaborators.**

| Gender diverse people | |
|---|---|
| **Inclusion criteria** | **Exclusion criteria** |
| • Aged over 18<br>• Fluent in German<br>• Identifies as gender diverse<br>• Sees or has seen a speech pathologist for gender affirming speech pathology care for a minimum of five training sessions | • Knows members of the research team before study participation |
| **Speech pathologists** | |
| **Inclusion criteria** | **Exclusion criteria** |
| • Aged over 18<br>• Fluent in German<br>• Qualified to practice as a speech pathologist<br>• Provides or has provided gender affirming speech pathology care in Germany | • Knows members of the research team before study participation<br>• Likely to be considered an expert by peer speech pathologists |
| **Advisory Group** | |
| **Inclusion criteria** | **Exclusion criteria** |
| • Aged over 18<br>• Fluent in German<br>• Identifies as gender diverse | • Participates in this study<br>• Knows members of the research team from a non-professional context |

pathologists who are widely known through their publications in gender affirming clinical practices. Excluding these clinicians has the disadvantage of missing potentially relevant insights on the research topic. However, this disadvantage is accepted in favour of avoiding potential conflicts of interest due to hierarchical power dynamics that may arise in the peer focus groups.

*AG collaborators.* Gender diverse people who know the members of the research team in a non-professional context will be excluded. People who know the team from a professional context can be included in the AG since they are collaborators in this project as opposed to study participants [74].

**2.4.2 Sampling.** Purposeful sampling will be used to include people who can provide rich information on the research topic and who can reflect on their everyday experiences [64, 75].

*Gender diverse people.* Apart from meeting the inclusion criteria, the goal for the sample of gender diverse study participants is to create variation within the sample, not only to counter the view that every gender diverse person's experience is the same [69], but also to acknowledge that people's experiences can differ depending on their gender identity and presentation [76–78]. Therefore, a group of gender diverse people who self-identify within a range of different gender categories and who have not all worked with the same speech pathologist will be included. Additionally, snowball sampling will be used for the gender diverse people, a strategy to contact people who may be more likely to participate in research with referral from others from their community [64, 79]. Gender diverse people may be particularly hard-to-reach due to potential stigmatisation they may have experienced in healthcare settings in the past [16]. If needed, gender diverse people who participate in the study will be asked to pass on the information about the study to others they know. A total of nine gender diverse study participants is planned to be included. This number is anticipated to be sufficient to support an in-depth understanding of the topic being researched in this study [80, 81]. The sample will be extended in case data analysis shows the need to gather more information for such an in-depth understanding [80].

*Speech pathologists.* For the sample of speech pathologists, purposeful sampling will be used [64, 75]. There are no further requirements other than meeting the inclusion criteria since the number of speech pathologists providing gender affirming care in Germany is estimated to be low. A total number of fifteen speech pathologists will be included for participation in three focus groups. Again, this number of focus groups is anticipated to be sufficient to support an in-depth understanding of this study's research topic [80, 81], but may be extended, if needed [80].

*AG collaborators.* For the AG, three people who identify as gender diverse will be included to collaborate in this research project and share their insights with the first author as members of the gender diverse community.

**2.4.3 Recruitment procedure and inclusion.** Different approaches will be taken to recruit gender diverse people, speech pathologists and AG members who are interested to participate or collaborate in this study.

*Gender diverse people.* To recruit gender diverse people, the following contacts will be approached: support groups and associations for gender diverse people in Germany, German speech pathologists who indicate on their webpage that they provide gender affirming care, and speech pathologists who provide gender affirming care and are known to the first author. These contacts will be asked to forward relevant study information to potential gender diverse study participants, e.g., via email, social media, or personal contacts.

*Speech pathologists.* The following contacts will be approached to recruit speech pathologists: German speech pathologists who are listed on webpages of vocational schools or universities as professionals teaching gender affirming care, who indicate on their webpage that they

provide gender affirming care, who are recommended as clinicians on webpages of the queer community, and who provide gender affirming care and are known to the first author. Speech pathologists known to the research team will not be included in the study but will be asked to pass on relevant study information to potential study participants. SP contacts will receive a brief overview of relevant study information via email.

*AG collaborators.* To reach out to potential collaborators, the same recruitment process as for the gender diverse study participants will apply in addition to contacting gender diverse people known to members of the research team personally. Gender diverse people known to the team in a non-professional context will not be included but will be asked to pass on study information to potential collaborators. Potential AG members will receive an email that includes a brief overview of relevant study information.

Since this study is conducted exclusively online, study participants and collaborators located in different geographical areas in Germany can be included [82]. People who are interested in being part of this project will be able to contact the first author, who will assess each person's eligibility based on the inclusion and exclusion criteria. If eligible, each person will receive comprehensive written information about the study and a consent form. They will have at least one week to read these materials before giving their written consent and commencing data collection. Study participation and collaboration will be voluntary. All study participants and collaborators will receive a small financial compensation for their time commitment following current ethical research recommendations [83].

## 2.5 Data collection

Data collection will take place exclusively online. This may impact study participants' level of engagement during data collection [82], but is overall not considered inferior to in-person data collection [84]. The first author will meet with each gender diverse person and each speech pathologist twice during data collection: once individually in a one-on-one videocall for a preliminary meeting and a second time in an individual videocall for the interview (gender diverse people) or a group videocall for the focus group discussion (speech pathologists).

**2.5.1 Preliminary meetings with study participants.** The initial, preliminary meeting with each study participant will be audio-recorded so that recordings can be used to prepare for the interviews with the gender diverse study participants, if needed. Technical issues with the videocall platform will be identified and resolved to prevent future problems. The contents of the written information and consent form will be explained, and the study participant's remaining questions clarified. Gender diverse study participants will receive a detailed verbal explanation of the interview process during the preliminary meeting to ensure they feel comfortable talking about personal topics during the interview. Since speech pathologists are considered less vulnerable than gender diverse people, they will receive a general explanation of the focus group process during the preliminary meeting and then a detailed explanation just before the start of the focus group discussion.

Relevant sociodemographic data will be collected from each study participant to help contextualise each study participant's experiences and perspectives in relation to the study aims. For the gender diverse study participants, this includes, e.g., their gender identity and the number of gender affirming SP sessions attended. For the speech pathologists, this includes, e.g., their SP educational background and their clinical work setting (see S3 Appendix for the comprehensive list).

To create a safe space, the study participants and the first author will get to know and become comfortable with each other, e.g., by talking about their profession or leisure activities during the preliminary meeting. A safe space can be particularly important for those study

participants who may have experienced gender minority stress in healthcare settings [16]. For example, some gender diverse study participants may have experienced situations wherein others did not respect their preferred name or pronouns [1, 85, 86].

The first, preliminary meeting and the second meeting for the interview or focus group discussion will take place anywhere from a few days to a few weeks apart, depending on each person's availability. Each study participant will receive an individual email with a summary of the most important information covered in the preliminary meeting to ensure they are sufficiently informed about the next steps in the study. This could include, for example, solutions to technical issues and answers to the study participant's questions.

**2.5.2 Setting of the interviews and focus group discussions.** Each interview and focus group discussion will take place online via videocall and will be audio-recorded. Interviews will take place in a one-on-one setting with the first author and focus group discussions in a group setting, led by the first author. Study participants will not have to answer interview questions or react to discussion prompts they feel uncomfortable with. They will be able to pause or end the interview or discussion at any time. The first author will take notes during the interviews and discussions, e.g., about participants' choice of wording and their non-verbal communication, e.g., if participants nod or use other gestures instead of words. The study participants' specific wording will be used when articulating questions during each interview and discussion as a way of creating a collaborative environment [87]. Noted non-verbal communication will be used as supporting information when creating transcripts of the verbal data.

The interviews and discussions are estimated to take approx. 90 minutes. Additional time will be scheduled to manage technical issues, clarify questions, and debrief. Debriefing will include a verbal summary of the interview or focus group discussion to which the study participants can add further thoughts they consider important. The gender diverse study participants will be made aware of different options of support in case of emotional distress, i.e., contacting the first author and local helplines.

After each interview and discussion, the first author will take further notes [88]. This may include, for example, her reflections about her behaviour as interviewer or facilitator, impressions of the interaction between the first author and the study participants including potential influences of the online format of data collection, and initial thoughts about conversation contents. The notes will serve as a basis to reflect on the data collection process with the research team and as a starting point for later data analysis.

**2.5.3 Episodic interviews with gender diverse people.** Gender diverse people will participate in one-on-one episodic interviews [66]. As opposed to group settings, a one-on-one setting can help to create a space in which the study participants feel comfortable to talk about sensitive topics such as PWB and identity [89]. Episodic interviews are based on the theoretical assumption that there are two types of knowledge: explicit, abstract knowledge and implicit, episodic knowledge [66]. While abstract knowledge is partly derived from experiences, but independent of concrete situations [88, 90], episodic knowledge is knowledge about concrete situations (episodes), closely connected to people's everyday experiences. The advantage of targeting episodic knowledge in interviews is that it provides insight into people's meaning-making in their day-to-day life [88, 91]. Episodic interviews access episodic knowledge by prompting narratives about concrete situations in the interviewee's life [66, 91]. In episodic interviewing, narratives are considered to comprise both a story (i.e., the sequence of events) and a plot (i.e., the speaker's attitudes, perceptions, and motives) related to an episode [91]. In contrast to episodic knowledge, abstract knowledge is accessed by asking general questions that evoke answers, e.g., about the interviewee's subjective definition of phenomena [66, 91]. Combining these prompts for both abstract and episodic knowledge will guide the interviewee

through the interview [66], while enabling them to describe events with their own emphasis and choice of wording [91]. The episodic interview is an interview form that is particularly open to both negative and positive narrations of situations [88].

An interview guide has been developed to prepare for the episodic interviews. It is based on the study aims and the research team's understanding of the current literature around PWB, identity, gender minority stress, (gender) self-presentation [13, 15, 17, 19, 21, 22, 56–59], and gender affirming SP practices [2, 27, 28, 42]. The interview guide contains questions and prompts around five different topics: (1) identity and PWB, (2) self-presentation, identity and PWB, (3) vocal patterns, identity and PWB, (4) experiences with SP practices, and (5) effects of SP practices. The first three topics aim to evoke data to explore the first research question, whereas the fourth and fifth topic are related to the second research question of this study (see S4 Appendix for the full episodic interview guide).

To minimise emotional distress, interview topics cover both negative and positive experiences with opportunities for the interviewee to guide the interview content in their preferred direction. However, the interview will still be open for new topics the interviewee wishes to raise [66]. The wording of the interview questions is based on the principle of opening up possibilities in conversations rather than narrowing them down, a common principle in narrative practices [46, 92]. For example, asking leading questions will be consciously avoided. Instead, the interviewee will be asked open questions so that they do not feel like they have to respond in any particular way (e.g., "Did you get the support through speech pathology that you wanted or was there something missing?"). In this way, the interview is designed to be as open as possible. Another example of this openness is that the interviewee will be invited to talk about their overall wellbeing instead of the narrower topic of PWB. This is in line with advice from the AG that highlighted that using the term 'psychological' in the context of the study could be perceived as pathologizing by some gender diverse people.

The interview guide has been reviewed by the AG and piloted with one cisgender person and two gender diverse people. After this process, the guide was finalised for data collection (see Table 2 for interview topics and example questions).

**2.5.4 Focus group discussions with speech pathologists.** Speech pathologists will participate in focus group discussions [67] to evoke both diversity and agreement about the research

**Table 2. Topics and example questions for the episodic interviews with the gender diverse people after piloting, revision, and modification.**

| Topics | Example interview questions[1] |
|---|---|
| Identity & (psychological) wellbeing | 1. What does "wellbeing" mean to you personally? |
| Self-presentation, identity & (psychological) wellbeing | * 5. When you think back: Do you remember a concrete situation in which it was particularly important or, vice versa, not important at all to present yourself through… [aspects previously named by interviewee]? Please tell me about it. |
| Vocal patterns, identity & (psychological) wellbeing | * 8. Please tell me about a situation where you felt uncomfortable with your voice [or name other aspects previously mentioned by the interviewee]. |
| | * 9. And vice versa? Please tell me about a situation wherein you felt comfortable with your voice [or aspects previously named by interviewee]. |
| Experiences with speech pathology practices | 18. If you think back and think of today, what do you think: Did you get the support through speech pathology that you wanted or was there something missing? |
| Effects of speech pathology practices | 23. Do you think speech pathology has changed something about how you see yourself or not? |

[1] Note: Questions evoking narrations are marked with an asterisk.

topics [93]. As opposed to one-on-one settings, the group process can support participants to explore their perspectives on a certain topic [94]. This is considered particularly helpful when reflecting on topics that are still rarely discussed in SP clinical literature, such as the speech pathologists' actions in supporting gender diverse clients' PWB and identity. Three focus group discussions will be conducted with four to five participants each, which is considered a rather small group size [93–95]. Small groups are chosen to make it more likely for every participant to engage in the online discussion. Further, participants who are assumed to be close confidants of each other, e.g., through being co-workers, will not participate in the same focus group discussion to encourage diversity of perspectives. However, for practical reasons, the final composition of each focus group will mainly depend on the study participants' availability.

A discussion guide has been developed to prepare for the focus group discussions. Again, it is based on the study aims and the research team's understanding of the current literature around PWB, identity, gender minority stress [13, 15, 17, 19, 21, 22, 56–59], and SP practices in gender affirming care [4, 29, 35, 42]. The focus group discussion guide contains questions and prompts around three different topics to evoke data to explore the third and fourth research question. The topics are: (1) General clinical work with gender diverse clients, (2) SP practices and identity, and (3) SP practices and PWB (see S5 Appendix for the full focus group discussion guide).

The discussion guide generally moves from broad to more specific questions around topics [67], such as SP approaches in gender affirming care or addressing identity in a SP context. Topics and their order will be partially flexible depending on the topics raised by the participants and the flow of the discussion [93].

The focus group discussion guide has been reviewed by the AG and piloted with one group of speech pathologists. It was refined and then presented to another speech pathologist working in gender affirming care. The speech pathologist was asked to think aloud [96], i.e., to verbalise their thoughts while reacting to the focus group discussion prompts. After this process, the focus group discussion guide was finalised (see Table 3 for focus group discussion topics and example prompts).

## 2.6 Data analysis and synthesis

Data from the episodic interviews with the gender diverse people and from the focus group discussions with the speech pathologists will be first analysed separately for each group of

**Table 3. Topics and example prompts for the focus group discussions with the speech pathologists after piloting, revision, and modification.**

| Topics | Example discussion prompts |
|---|---|
| General clinical work with gender diverse clients | 3. What is particularly important to you in your work in gender affirming care? Please share your thoughts with each other. |
| Speech pathology practices & identity | 6. [The following phrasing depends on the previous discussion.] (Some of) You agreed that self-image[1] or how your client experiences their own identity plays a role for their wellbeing. We know that wellbeing is a goal of healthcare for gender diverse clients. Does this mean that speech pathologists should work with gender diverse clients on their self-image or identity, or who would be responsible for this? |
| Speech pathology practices & wellbeing | 10. Let's take a look at wellbeing. Would you say that you work on your clients' wellbeing or not? If yes, how do you do that? |

[1] Note: The term 'self-image' is used in a lay manner as a synonym for 'identity' to give the focus group participants an idea of how identity is understood.

study participants using RTA [52]. Findings from each group will then be compared through data source triangulation [97, 98].

**2.6.1 Analysis of episodic interviews and focus group discussions.** Transcripts of each interview and focus group discussion will be created as a basis for data analysis. Initial transcripts will be created using an automated speech recognition service [99] and then reviewed manually following transcription guidelines by Braun and Clarke [100] and Kuckartz and Rädiker [101, 102]. Since these existing guidelines were not considered sufficiently detailed for this study, they were further specified by the research team (e.g., how to note overlapping speech).

A variation of Baun and Clarke's RTA [52] will be used for the analysis of the interviews and of the focus group discussions. The aim of RTA is to develop patterns of meaning, that is, themes that are relevant for the research questions [52]. Themes will not be created a priori. Instead, the data analysis process will be data driven and the first author will actively construct themes [52, 103, 104]. As a core element of RTA, generating themes will be shaped by the first author's constant reflections about the context in which the analysis takes place (see section 2.1 for a reflexivity statement) [52, 103, 104]. Both semantic and latent meaning will be explored with RTA. This will include surface-level meaning that remains closer to the study participants' reports, as well as underlying, more implicit meaning within the data that is more influenced by the first author's active endeavours to create meaning [52, 104].

To guide the process of developing themes, RTA encompasses six phases which build on each other [52]. After familiarisation with the data by reflecting and taking notes on each interview and focus group discussion in phase one, data will be explored more systematically in phase two. Codes that contain ideas or meaning will be created based on segments of the collected data relevant to the study aims. In phase three, codes will be clustered together to generate initial themes that encompass a similar idea or meaning. These themes will be further refined, merged, or separated in phase four. This phase will result in a final thematic map. Themes will be named and defined in phase five. Definitions will contain the scope, boundaries, and concept of each theme. Phase six will involve writing a report with a description of the analytic process. This is part of the analysis itself, wherein the results of the analysis will be further refined and shaped [52]. Generally, the RTA process is iterative and therefore will involve going back to earlier points of the analysis to re-analyse data with new insights [52, 105].

The RTA approach will be expanded by using various strategies throughout different phases of the analysis, including memos [106], member reflections [107, 108], and peer debriefing [109]. The aim of these strategies is to facilitate further engagement with the data and to support the first author to construct meaningful themes [52]. Table 4 shows how and when the different strategies will be used.

Data, all of which is in German, will be primarily analysed by the first author. Member reflections [107, 108] with the study participants and peer debriefings [109] with the research team, with peers who are qualitative researchers, and with the AG collaborators will be used as important strategies to acknowledge diverse perspectives in the data analysis process and to facilitate the first author's deeper understanding and engagement with the data [52, 107].

Transcripts, which are in German, will be analysed in German in phases one to four of RTA to avoid potential language restrictions in English. Supporting strategies, i.e., member reflections and peer debriefings will be performed in German with study participants, collaborators, peer researchers, and the German-speaking part of the PhD research team (JT, HH). Whenever additional support is needed in the data analysis process, parts of the data will be translated into English for additional discussions with the whole PhD research team (JT, JO, SQ, HH). At the end of phase four, themes will be translated to perform phase five and six of the analysis in English and to subsequently discuss data with the research team.

**Table 4. Complementary strategies to facilitate analysis and engagement with the data during the different phases of reflexive thematic analysis [52].**

| Strategies | Implementation of strategies | Phase of analysis |
|---|---|---|
| Memos [106] | For an in-depth engagement with the data: Write down thoughts on<br>• prompts stimulating general reflections [52, 106];<br>• prompts stimulating reflections specific to the research aims [52];<br>If a more in-depth engagement with participants' stories is needed: Write down thoughts on questions to engage with the narrative parts (story and plot of an episode) of the data [based on: Strauss & Corbin, 1990, pp. 27–28, as cited in 66, 110]. | Phase one to five |
| Member reflections [107, 108] | Send study participants a summary of the first author's ideas about potential initial themes constructed from the interviews or focus group discussions. Ask study participants to share their reflections on these ideas. Reflect on each study participant's feedback. Study participants can contact the first author in case they experience distress anytime during this process. | Phase three |
| Peer debriefing [109] | Discuss parts of the data within the research team, with peer qualitative researchers, and the AG collaborators. | Phase one to five |

The first author will constantly document her critical reflections about the study in a reflexive journal during data analysis [52, 103]. Reflections include thoughts on how this research project is shaped by the members of research team, their background, assumptions, values, and methodological choices [52, 103]. This is particularly important since the first author, even though supported by others, will perform data analysis mainly by herself [52]. This strategy is known as researcher reflexivity and is a crucial part of RTA [52].

The software NVivo [111] will be used for data management during the analysis process.

**2.6.2 Synthesis of interview and focus group findings.** After data has been analysed for both the episodic interviews and the focus group discussions, the findings from both data sources will be synthesized using data source triangulation [97, 98]. This method will help to gain a more comprehensive and in-depth understanding of the phenomena under research [97, 98], i.e., PWB and identity in a gender affirming SP clinical context. Data synthesis will be implemented in a way that is coherent with the previous data analysis process and the underlying theoretical assumptions of this study. For example, the Triangulation Protocol for Qualitative Health Research [112] may be used to compare and contrast the perspectives of gender diverse people and speech pathologists on the research topic.

## 3 Collaboration with the gender diverse community

The gender diverse community is actively involved as collaborator in this study [113] to design and conduct research that is likely to be appropriate and relevant to the gender diverse community [27]. This is particularly important since the majority of the research team has no lived experience of identifying as gender diverse.

Three gender diverse people form an AG to provide advice throughout the study based on their insights and experience of identifying as gender diverse. Consultations include, for example, feedback on the content and appropriateness of wording of the interview and focus group discussion guides, reflections during data analysis, and ideas of how to accessibly disseminate study results to the gender diverse community. The AG members either provide written feedback or verbal advice during regular group videocalls. In-between consultations, the AG members are kept up to date about the study progress via email or group videocalls. An overview of the AG collaborators' contributions in this study is provided in the supporting information (see S6 Appendix).

A memorandum of understanding has been created by the research team to communicate transparently mutual expectations from the research team and the AG members [113]. It covers, for example, agreements about mechanisms for preserving members' confidentiality, the nature of the contributions of AG members and the research team, financial compensation, and acknowledgements on research outputs. The memorandum of understanding was reviewed by the AG before its approval.

To share how different parties contributed to this study in a standardised and transparent way, the data sharing system Standardised Data on Initiatives (STARDIT) [114] is used and updated repeatedly.

## 4 Ethical principles

General ethical principles apply to this study to ensure no harm to study participants, collaborators, the gender diverse community, and the community of SP [115]. This is particularly important when working with people from vulnerable or marginalised communities [69, 116], which is why recommendations for research with gender diverse people [69, 116] and for trauma-informed research [89] apply. This includes, for example, using appropriate language, informed consent, free choice of pseudonyms, financial compensation, and ensuring data privacy. A more detailed implementation of these recommendations is described in different sections of this protocol.

Hardcopy and electronic data, including audio files and transcripts, will be stored securely at La Trobe University. Participant data will remain confidential throughout and after the study. No identifying information about the study participants will be revealed at any time.

## 5 Research quality

The study is based on current recommendations of quality criteria in qualitative research [109, 117, 118] and criteria specific for RTA [52, 53].

To ensure a worthy topic [117] and therefore to conduct research that is relevant, the research topic is based on current transdisciplinary literature and the research team's individual clinical experience [117]. The collaboration with the AG ensures that the topic is relevant to the gender diverse community. The study is designed so that it acknowledges a rich variety of theories and data sources, which allows for a nuanced and complex engagement with the research topic [117]. Further, to ensure coherence [52, 80, 117], so that the research aims align with the methodology and approaches for data collection, for data analysis, and for reporting study findings, literature was reviewed thoroughly and the different parts of the study design were repeatedly discussed within the research team. Credibility [109, 117, 118], or the plausibility of research findings, will be reached through reflexivity throughout the whole study and through using member reflections and peer debriefing to gain more insight into the data. Credibility is further supported by contextualising the study findings. This means providing great detail on the study context (e.g., about the research team and study participants) to transparently show how the study findings are generated [52]. Reflexivity and creating an audit trail, in which decision making is described and justified will be used to meet the criteria of dependability [109, 118] and confirmability [109, 118]. The goal is to make the research process transparent (dependability) and to show that the data represents the phenomenon that is researched (confirmability). To contribute to transferability [109, 117], the degree to which qualitative research findings can be transferred and therefore be valuable for other groups or contexts, two strategies will be performed: contextualisation of study findings, and purposeful sampling to achieve variability in the sample of study participants [118]. Overall, by pursuing reflexivity and being transparent through strategies including audit trails and contextualisation of study results, the criterion of sincerity [117] will be met.

## 6 Dissemination

The research process and the research outcomes will be communicated and disseminated following the RTARG [53]. The target audience are healthcare providers and researchers working in gender affirming care, primarily speech pathologists, as well as the gender diverse community itself. Study results will be disseminated to professionals through the doctoral thesis, publications in journals, and conference presentations. Publications are planned to be open access whenever possible. The strategy for disseminating the study results to the gender diverse community will be developed in close collaboration with the AG [116] to ensure accessibility to the study results [119]. Dissemination to the gender diverse community could include, for example, information via email, such as plain language summaries (see S7 Appendix), and presentations in support group meetings or community conferences. Before the dissemination of the final report, each study participant will be contacted to confirm they feel comfortable with the presentation of their data, such as the chosen example quotes from the interview or focus group discussion.

## 7 Discussion

Supporting gender diverse clients' PWB is one key goal of gender affirming healthcare [1]. Literature shows that experiencing identity as positive is crucial for a person's PWB [13, 58], but that developing a positive identity story can be challenging for gender diverse people [15]. Bringing together theories about PWB [13, 58], gender minority stress [15, 56], narrative identity [21, 22], and identity development in gender diverse people [17] could help determine what role these concepts that centre identity may play for a gender diverse person's PWB in a SP context.

PWB and identity beyond gender are rarely discussed in a gender affirming SP context. Providing holistic care and supporting clients' PWB requires an open mindset from speech pathologists regarding how they see their clients and their own role as clinicians [23]. The present protocol describes a qualitative study that aims to shed light on PWB and identity in a gender affirming SP context, both from gender diverse people's and speech pathologists' perspectives and experiences with the research topics.

This study protocol is important to meet quality criteria of qualitative research, such as credibility and transferability of the study [109, 117, 118]. Accordingly, the protocol provides great detail on the planned study, including the research team's reflexivity statement, a description of the context, and different steps planned for this study and outlines potential influences on the research process.

The first author's limited experience in qualitative research, particularly in qualitative data analysis, could be seen as a limitation to the study. However, strategies such as reflexivity [52], member reflections [107, 108], peer debriefing [109], and guidance from the research team will facilitate the first author's data analysis process.

As SP is just one component that may be included in gender affirming healthcare [1], the focus of the planned study on PWB and identity in a SP context is a potential limitation. Therefore, future studies around PWB and identity in gender affirming care could centre disciplines beyond SP, such as, psychology, endocrinology, or occupational therapy to gain insights on the research topic from a different perspective and in a different context. Existing research in psychology on PWB and identity in gender diverse people [e.g., 19, 45] could be a valuable resource when planning such studies.

Moreover, the information about the study shared in the recruitment process may influence who expresses interest in participating in the study. For instance, gender diverse people and speech pathologists who consider PWB and identity important in gender affirming SP

practices may be more willing to participate in the study than people who hold a different belief. Furthermore, gender diverse people who have had experiences of discrimination or harassment in healthcare settings in the past [10, 16] may not want to participate in a study conducted by a healthcare professional, such as a speech pathologist. This could limit the diversity of perspectives on the study topic and could be the subject of future research projects. However, the purposeful sampling strategy in this study ensures that study participants will be able to provide rich information on the study topic [64, 75]. Purposefully aiming for a variety in the sample of gender diverse people will contribute to a diversity of perspectives and experiences and therefore to transferability of study findings [118].

The online setting of the study limits the study participants to people with access to the internet. It could also inhibit the study participants' capacity for in-depth engagement in the interview or discussion due to potential distractions from their surroundings [82]. However, online data collection is not considered inferior to in-person data collection [84]. Moreover, it allows inclusion of participants located in different geographical areas [82].

Another limitation is that the study participants will meet the first author only once before participating in an interview or focus group discussion. For some participants, it may therefore be challenging to share their perspectives and experiences about sensitive topics such as their PWB or identity. However, a clearly defined professional setting with professional distance may also be helpful for some participants. In general, making the study participants feel as comfortable as possible by creating a setting that feels safe [89] could assist study participants to share the information they would like to.

The research team chose the data collection methods for this study due to their individual advantages as outlined further above in this paper. Future studies could, of course, choose a different study design. For example, conducting one-on-one interviews with speech pathologists would make it feasible to include experts in the field without the risk of hierarchical power dynamics arising within peer focus groups.

The synthesis of the findings of both the episodic interviews and the focus group discussions brings together the perspectives and experiences of gender diverse people and speech pathologists on the research topic. This is a valuable opportunity to gain insights into similarities and differences regarding the needs of these two groups. These insights may be useful for developing strategies to improve SP services for gender diverse people. Based on the findings from this qualitative study and other research in the area of gender diversity, PWB, identity, and SP, future research could apply both qualitative and quantitative methods to further explore the research topic. For instance, results from research using standardised tools to investigate gender diverse people's PWB and experienced identity could be later used to develop a holistic assessment in gender affirming SP care.

In summary, the researchers anticipate that the study findings will contribute to further understanding gender diverse people's PWB and its connection with identity in the gender affirming SP space. Linking the study findings to the emerging discussion in SP around PWB could help to progress the discourse in gender affirming SP practices beyond approaches around gender and vocal patterns. The study findings could contribute to diversifying gender affirming SP practices to tailor them to the unique needs of gender diverse clients.

## Supporting information

**S1 Appendix. Reflexive Thematic Analysis Reporting Guidelines (RTARG) checklist filled in for the planned study.**
(PDF)

**S2 Appendix. Online platforms and software for different phases of the study to maintain study participant data privacy.**
(PDF)

**S3 Appendix. Sociodemographic data to collect from participating gender diverse people and speech pathologists.**
(PDF)

**S4 Appendix. Guide for episodic interviews with gender diverse study participants.**
(PDF)

**S5 Appendix. Guide for focus group discussions with participating speech pathologists.**
(PDF)

**S6 Appendix. Contributions of the Advisory Group and research team during the different phases of the study.**
(PDF)

**S7 Appendix. Plain language summary.**
(PDF)

## Acknowledgments

The authors would like to thank the members of the Advisory Group, Tommy Balaz, Luna Dremmen and Patricia Schüttler (named with their consent), for their contribution and for sharing their valuable insights as members of the gender diverse community. The authors would also like to thank the PLOS ONE reviewers for their constructive and thought-provoking feedback on this article.

## Author Contributions

**Conceptualization:** Julia Tanase, Sterling Quinn, Sabine Corsten, Jennifer Oates.

**Investigation:** Julia Tanase.

**Methodology:** Julia Tanase, Sterling Quinn, Sabine Corsten, Hilke Hansen, Jennifer Oates.

**Supervision:** Sterling Quinn, Hilke Hansen, Jennifer Oates.

**Writing – original draft:** Julia Tanase.

**Writing – review & editing:** Julia Tanase, Sterling Quinn, Sabine Corsten, Hilke Hansen, Jennifer Oates.

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
