## [Decision Letter · Decision Letter 0]

15 May 2024

PONE-D-23-43876Gender diverse people’s psychological wellbeing and identity in the context of gender affirming speech pathology practice: A qualitative study protocolPLOS ONE

Dear Dr. Tanase,

Thank you for submitting your manuscript to PLOS ONE. After careful consideration, we feel that it has merit but does not fully meet PLOS ONE’s publication criteria as it currently stands. Therefore, we invite you to submit a revised version of the manuscript that addresses the points raised during the review process.

We look forward to receiving your revised manuscript.

Kind regards,

Vanessa Carels

Staff Editor

PLOS ONE

Journal Requirements:

Reviewers' comments:

Reviewer's Responses to Questions

**Comments to the Author**

1. Does the manuscript provide a valid rationale for the proposed study, with clearly identified and justified research questions?

Reviewer #1: Yes

Reviewer #2: Yes

Reviewer #3: Partly

2. Is the protocol technically sound and planned in a manner that will lead to a meaningful outcome and allow testing the stated hypotheses?

Reviewer #1: Yes

Reviewer #2: Yes

Reviewer #3: Partly

3. Is the methodology feasible and described in sufficient detail to allow the work to be replicable?

Reviewer #1: No

Reviewer #2: Yes

Reviewer #3: Yes

4. Have the authors described where all data underlying the findings will be made available when the study is complete?

Reviewer #1: Yes

Reviewer #2: Yes

Reviewer #3: Yes

5. Is the manuscript presented in an intelligible fashion and written in standard English?

Reviewer #1: Yes

Reviewer #2: Yes

Reviewer #3: Yes

6. Review Comments to the Author

You may also provide optional suggestions and comments to authors that they might find helpful in planning their study.

Reviewer #1: Comments to the authors.

Thank you for this well-planned study protocol which is supposed to be used to increase knowledge about trans peoples´ well-being in the context of speech pathology practice for gender-affirming voice and communication training. The manuscript is well-written. The purpose of the planned study is clear and will fill an evident knowledge gap. Only a few studies have been performed in the area previously.

Abstract – fine!

Background – the references referred to are relevant and up-dated and well described.

Page 3, line 51- Please define the term cisgender

Page 4, line 72. Please expand on the meaning of ‘identity fluid’ so that the concept is not misinterpreted as ‘gender fluid’.

Page 5. Lines 120-122. This question may also be related to the context which the SP work within. In many European countries, the SP is part of an inter-professional gender team (including psychiatrists and psychologists) which together works on a ‘holistic’ approach. This is probably quite different from other clinical contexts such as private practicing SPs.

Page 6. Lines 127-128. Based on the SP context the first question could be addressed in a broader sense to the gender team, and not only to the SP.

Page 6. Lines 135-137. “The research questions are based on the research team’s current knowledge”. Then it is important to understand a bit more about the research team’s clinical context (private, hospital, part of a gender team) and education?

Methods:

At this stage the study is not yet completed, and you still work on the procedure for the data collection and analysis. Therefore, it would not be possible to replicate the study based on the study protocol.

How many participants (clients and SPs) do you plan to include?

Page 7, line 151. Specify the ‘gender affirming SP contexts’.

Page 7. Lines 154-155. Please motivate why you will use different methods for collecting data: individual interviews for clients and focus groups for SPs.

It seems good and meaningful to collect data in two parts to be able to stop and reflect on the data collecting process before continuing the full study.

Page 7, line 177. A . I missing after practices.

Page 8, line 188. You have a hypothesis “that speech pathologists working in this area might find it challenging to think of themselves as clinicians supporting clients’ PWB”. Please describe what makes supporting transgender clients in their PWB so different from supporting other clients in the SP care in their PWB (such as clients with cancer, neurodegenerative diseases etc). Do not most SPs work with councelling to give psychological support to patients with communication disorders? SPs must be well-prepared but need special education about transgender clients and minority stress of course if they are not experienced with the client group.

Page 8. Line 196. Please reflect on the advantages and disadvantages with on-line interviews.

Page 9. You have decided to exclude SPs who are “widely known” to avoid hierarchical power dynamics in the focus groups. If those SPs are also highly experienced in the area of gender affirming SP practices, are you not worried to miss important from those? Why do you not offer individual interviews also with the SPs as with the clients? And what is the definition of “widely known” and who will decide this? There seems to be a risk of bias if this is the investigator’s decision.

Page 11, line 250. So, recruitment is already completed? Later it says that data collection is completed, confusing.

Page 11, line 251 speech pathologists - SP. Earlier in the text you have written the abbreviation SP and later speech pathologist. Please be consistent.

Page 12, line 264. You write that the SPs “are known to the investigator”. Isn’t this a risk of bias?

Page 12, Line 267. This is also written for the AG group “known to the investigator personally”. Risk of bias?

Page 12-13. Line 281-284. It is somewhat unclear if the investigator will meet the SPs individually before the focus group.

Page 13, line 294-295. “Speech pathologists will receive an explanation of the focus group process just before the start of the focus group discussion”. Unclear if this is taken place in the individual meeting with the SP.

Page 13, lines 296-207 “providing gender affirming SP services”. What more information will you collect from the SPs? I would assume that information about working environment (working alone, working with colleagues, with a interprofessional gender team), years working with transgender clients, with patients with voice disorders, and what kind of diagnoses. What kind of education do the SPs have, BA? MSc? Advanced courses in SP? Have they been participating in national/international conferences within the transgender area? This is important to describe since SP educations in Europe differ a lot (some are even not at a university level). This is important if ou think that your results can be generalized.

Page 13, line 303. A . is missing after the parenthesis.

Page 14. 315-317 “Notes will be taken during data collection, e.g., about participants’ choice of wording or their non-verbal communication”. Please clarify if this will be done from the audio-recordings after the sessions, or from video-recordings that seems necessary if non-verbal communication will be described. It is written earlier that audio-recordings are carried out. Please clarify.

Page 14, line 328. “..reflections about the investigator’s behaviour”. Who will do those reflections?

Table 2. Items 15 and 18 are leading questions (which you wrote earlier in the text that should be avoided).

Status of the project

Page 23, line 517: Data collection is planned to be completed by 29 February 2024. Please update this sentence. Did you already complete data collection? On page 11 you write that “recruitment period is anticipated take place from 1 June 2023 to 29 February 2024”. Recruitment and data collection is not the same.

Discussion

You have lifted relevant issues regarding methodological issues and limitations. Maybe add a sentence about if you think that you results can be generalized.

References:

Check consistency regarding capital letters (or not) in the titles of the articles.

Ref 10 and 17 – are there more information?

The results from this study, when finished, will be useful for many SPs working in similar clinical contexts as the one in the study. I widh you the best for completing the study.

Reviewer #2: Is it ok for a registered report to be published AFTER data collection? Line 250 says recruitment takes place June 2023 to February 2024.

The proposed qualitative study is justified and well-designed. It is scientifically rigorous and ethically conscientious.

The methodology is feasible and detailed, although without all the interview questions it cannot be replicated as well. Could the full list of questions be provided as supplemental material?

Without the German transcripts being made available, it may be wise to have more than 2 people conducting the analysis part of this study. As diverse as the research team is, the meaning-making at this stage is critical and limited by who is able to participate in it.

Four research questions are provided but the hypotheses are not clear enough to be tested. Due to the nature of this exploratory qualitative study, the hypotheses do not need to be testable. However, it would be helpful to explicitly state the implied hypotheses or assumptions found in the research questions in order to justify the line of questioning and build the results into a testable hypothesis or theory. This seems feasible based on the provided background information. My understanding of the rationale would suggest that the authors hypothesize that…

1. gender-diverse people seeking SLP support (can/can’t) experience their identity as positive. Those experiencing it as positive are more likely to include sources of support in their stories.

2. gender-diverse people seeking SLP support (do/don’t) identify changing vocal patterns AND managing their distress regarding communication as valuable. They identify misgendering and other factors in the minority stress model as not valuable.

3. speech pathologists (do/ don’t) see their role as supporting voice as well as PWB and perspective on identity.

4. speech pathologists report providing support ranging from … (referrals to mental health practitioners to providing counseling).

In other words, it is hypothesized that the outcome data will align with the empirical and best practice literature described in the introduction and provide further detail.

Reviewer #3: Thank you for the opportunity to review this study protocol.

Study protocols for qualitative research projects are as important to publish as those of quantitative research efforts. However, the dynamic approach to information gathering used in the qualitative research method makes a pre-hoc description of the study design a challenge.

This particular study focuses on the psychological wellbeing and identity of clients of gender affirming voice training and have opted to use a reflective approach to analysis of the material. Community participation is built in in the planned research. I have a couple of reflections that I would like the authors to consider before starting the research.

First, the study focuses on the clients' identity and wellbeing, and while the study seems well equipped to provide a well balances presentation of the clients' identity, I am not sure that wellbeing is captured adequately. At least not all applicable aspects of wellbeing I can think of. With an answer to the question of "how" the study protocol will allow elucidation of the study aims, I cannot se a strong reason why it should be published as a study protocol. As it may not be of use to others (the readers) wanting to throw light on the same or related aspects in a different context. The authors need to strengthen the link between wellbeing, according to all applicable definitions, and the planned discussions with participants, in particular with the transgender participants. Please also note that there are recent works that you have not included in the preparation of the study protocols that brings forward the fears of transgender clients when entering into training

Holmberg, J., Linander, I., Södersten, M. & Karlsson, F. (2023). Exploring Motives and Perceived Barriers for Voice Modification: The Views of Transgender and Gender-Diverse Voice Clients. Journal of Speech, Language, and Hearing Research, 66(7), 2246–2259. https://doi.org/10.1044/2023_jslhr-23-00042

And a more expansive discussion of subgroups in the transgender speakers, with regards to gender identity, that the authors need to re-consider (while one is cited)

Azul, D. & Hancock, A. B. (2020). Who or what has the capacity to influence voice production? Development of a transdisciplinary theoretical approach to clinical practice addressing voice and the communication of speaker socio-cultural positioning. International Journal of Speech-Language Pathology, 22(5), 559–570. https://doi.org/10.1080/17549507.2019.1709544

Merritt, B. (2023). Speech beyond the binary: Some acoustic-phonetic and auditory-perceptual characteristics of non-binary speakers. JASA Express Letters, 3(3), 035206. https://doi.org/10.1121/10.0017642

In this work. How barriers to voice training comes into may interact with both the client's and SLPs views is not clearly addressed in the study protocol. And with wellbeing not being well defined, there is no strong presence of participant validation of transcripts or conclusions drawn, and there is not standardized tool used to further elucidate the level of stress perceived, the study protocol is too likely to not provide a strong basis for a description of high validity.

Please provide a stronger link between the research questions asked, the planned interview components of both the transgender and SPL participants, a plan for how the SPL and transgender findings will be contrasted or linked, and stronger processes aiming for a grounding the analyses with the participants and the broader community of clients.

7. PLOS authors have the option to publish the peer review history of their article (what does this mean?). If published, this will include your full peer review and any attached files.

Reviewer #1: No

Reviewer #2: No

Reviewer #3: No

---

## [Author Response · Author response to Decision Letter 0]

16 Jun 2024

Reviewer #1: Comments to the authors.

Thank you for this well-planned study protocol which is supposed to be used to increase knowledge about trans peoples´ well-being in the context of speech pathology practice for gender-affirming voice and communication training. The manuscript is well-written. The purpose of the planned study is clear and will fill an evident knowledge gap. Only a few studies have been performed in the area previously.

Abstract – fine!

Background – the references referred to are relevant and up-dated and well described. 

RESPONSE: Thank you for your thought-provoking feedback. We appreciate your thorough review of our manuscript and your valuable comments. You will find our reply to each of your comments below. The pages, sections, and lines we refer to in our replies are those in the revised unmarked manuscript version.

Page 3, line 51- Please define the term cisgender 

RESPONSE: We have provided a clearer explanation of this term when we describe the difference in depression, anxiety, etc. between gender diverse people and “cisgender people who do identify with their gender presumed at birth”, see line 53.

Page 4, line 72. Please expand on the meaning of ‘identity fluid’ so that the concept is not misinterpreted as ‘gender fluid’. 

RESPONSE: We have elaborated more on identity to avoid confusion and stated that “This ongoing, individual and interindividual meaning-making means that a person’s identity is in constant change and therefore mutable”, see lines 88-89.

Page 5. Lines 120-122. This question may also be related to the context which the SP work within. In many European countries, the SP is part of an inter-professional gender team (including psychiatrists and psychologists) which together works on a ‘holistic’ approach. This is probably quite different from other clinical contexts such as private practicing SPs. 

RESPONSE: We agree with the point you raise (referring to the original statement that “at present, it is unknown whether speech pathologists who provide gender affirming care view their role in this way and whether they already apply more holistic approaches to their clinical work.”). 

To support our study aims, we have added a new section to elaborate more on the clinical professionals and explain that how a clinician sees their role in supporting gender diverse people might depend on the context in which they practice their profession. We have given an example to explain that the clinical setting in which speech pathologists provide gender affirming care (i.e., interdisciplinary healthcare clinics vs. single discipline settings with peer speech pathologists) might influence whether they feel responsible for addressing client psychological wellbeing. We have also stated that it is unclear how the professional setting influences speech pathologists’ mindset regarding client support. The new section can be found in lines 148-158.

Page 6. Lines 127-128. Based on the SP context the first question could be addressed in a broader sense to the gender team, and not only to the SP. 

RESPONSE: We agree that it will be important to explore this question related to other professions in future research. However, due to the lack of research around identity and psychological wellbeing in gender affirming speech pathology clinical care, other professions are not our current focus. 

To address your point, we have stated that our research could be a good starting point for similar research around identity and psychological wellbeing in gender affirming healthcare, see lines 189-192. We have also suggested in our discussion that focussing on speech pathology in gender affirming healthcare is a limitation of the study and that studies with other professions are warranted to gain insights on the research topic from a different perspective and in a different context, see lines 701-705.

Page 6. Lines 135-137. “The research questions are based on the research team’s current knowledge”. Then it is important to understand a bit more about the research team’s clinical context (private, hospital, part of a gender team) and education? 

RESPONSE: Thank you for this valid point. We agree that the information originally given in the reflexivity statement in one of the later sections, is not sufficient and that the reader needs more guidance and information here. To address this, we have:

- removed the sentence you are referring to in order to avoid confusion.

- given the reader more information about the background of the research team. We have added 

 - The clinical setting in which members of the research team provided gender affirming care, see lines 206-207 and 215-216.

 - The multidisciplinary educational background (speech pathology, sociology, psychology) and research interests (identity in speech pathology) of the different members of the research team, see lines 213-215.

Methods:

At this stage the study is not yet completed, and you still work on the procedure for the data collection and analysis. Therefore, it would not be possible to replicate the study based on the study protocol. 

RESPONSE: Thank you for the comment. As we are unsure as to what exactly you are referring to, we would like to address your comment in two different ways:

In case your feedback refers to ensuring that the study provides enough detail to be replicated: 

Even though the study has not been completed yet, we would like to provide as much detail as possible about the planned study via this study protocol. We do this to support others who would like to take a similar methodological approach and/or would like to use the theoretical conceptualisations of psychological wellbeing and identity for their own research in this area. We have added this as an aim of the study protocol, see lines 186-192. 

To provide more detail about our study, we have added the interview guide and focus group discussion guide as supporting information, see S4 Appendix and S5 Appendix. This has led to editing changes in the manuscript:

- We have referred to the full interview and focus group discussion guides as supporting information, see lines 460-461 and lines 502-503.

- To match the layout of the interview guide in the supporting information with the example questions in the manuscript, we have edited Table 2 in the manuscript, see p. 20, Table 2. We have not made any changes to the content of the table.

- We have listed the interview and focus group discussion guide as supporting information, see lines 767-769.

Further, we have explained the value of the study protocol in the discussion section. We have now stated that the study protocol is important to meet quality criteria of qualitative research (credibility, transferability) and that it provides great detail on the planned study, see lines 692-696.

In case your feedback refers to the fact that the study is not yet finished as qualitative research is a dynamic process: We have outlined the value of the study protocol and have stated that comparing the study protocol with later publications on the study will demonstrate the dynamic process of qualitative research, see lines 181-185.

How many participants (clients and SPs) do you plan to include? 

RESPONSE: We plan to include nine gender diverse people, and fifteen speech pathologists for three focus group discussions. We have now included this information in section 2.4.2 “Sampling”. We have also stated that we anticipate this number to be sufficient to support an in-depth understanding of the research topic, but that the sample will be extended in case data analysis shows the need to gather more information. We explain this for both the gender diverse people and the speech pathologists in lines 324-327 and 332-334 respectively. We plan to include three community advisory group members and now have added this information as well, see lines 335-337.

Page 7, line 151. Specify the ‘gender affirming SP contexts’. 

RESPONSE: Thank you for this. To highlight that we focus on speech pathologists who work clinically in the area of gender diversity we have:

- specified that “‘clinical’ refers to any healthcare setting providing direct client support, i.e., private practices, or healthcare clinics.”, see lines 246-247.

- added the word “clinical” when the context of gender affirming care might be unclear, see lines 170, 246. We also have added the word “clinical” for new writing added in the manuscript, whenever the context might be unclear. 

Page 7. Lines 154-155. Please motivate why you will use different methods for collecting data: individual interviews for clients and focus groups for SPs. 

RESPONSE: We agree that this is important information. We have given a short explanation in the section you are referring to (2.3 “Methodology and study design”). We have now stated that episodic interviews will be conducted to gain a detailed insight into gender diverse people’s personal experiences on the research topic and that focus group discussions with speech pathologists will be conducted to facilitate intense discussion about the research topic, see lines 252-255.

In the data collection sections (2.3.5 and 2.3.4), we have now also elaborated more clearly on why we chose each method.

For the episodic interviews, we now have explained that we chose this method 

- to create a safe space to talk about sensitive topics (psychological wellbeing, identity).

- to target both episodic and abstract knowledge. We have highlighted here that the advantage of targeting episodic knowledge in interviews is that it provides insight into people’s meaning-making in their day-to-day life.

- to have an interview form that is particularly open to both negative and positive narrations of situations.

We have explained this in lines 437-452.

For the focus group discussions, additional to the aim of evoking both diversity and agreement about the research topics, we now have contrasted that the group process can support participants to explore their perspectives on a certain topic as opposed to one-on-one settings. We have specified that speech pathologists’ actions in supporting gender diverse clients’ psychological wellbeing and identity are rarely discussed in speech pathology literature. We have explained this in lines 485-489. 

It seems good and meaningful to collect data in two parts to be able to stop and reflect on the data collecting process before continuing the full study. 

RESPONSE: Thank you for your kind feedback. 

Page 7, line 177. A . I missing after practices. 

RESPONSE: Thank you for noticing this. We have corrected the interpunction, see line 208.

Page 8, line 188. You have a hypothesis “that speech pathologists working in this area might find it challenging to think of themselves as clinicians supporting clients’ PWB”. Please describe what makes supporting transgender clients in their PWB so different from supporting other clients in the SP care in their PWB (such as clients with cancer, neurodegenerative diseases etc). Do not most SPs work with councelling to give psychological support to patients with communication disorders? SPs must be well-prepared but need special education about transgender clients and minority stress of course if they are not experienced with the client group. 

RESPONSE: Thank you for this valuable thought. We would like to reply to the different points you raise individually:

Supporting speech pathology clients through counselling:

We agree that counselling should be part of any speech pathology approach with clients. However, we found that this is not necessarily the case. We now have briefly outlined that some speech pathologists might be hesitant to use counselling in their work. Clinicians might be unsure about their scope of practice and might even feel conflicted when counselling clients, as shown in research in the field of aphasia rehabilitation, see lines 159-164.

Education about gender diverse clients and minority stress:

To address your thought, we now have stated that additional to professional training in gender affirming voice modification approaches, a full understanding of a gender diverse client’s individual challenges and resources in life is crucial when providing gender affirming care that aims to support client psychological wellbeing, see lines 121-123.

Page 8. Line 196. Please reflect on the advantages and disadvantages with on-line interviews. 

RESPONSE: Thank you for this comment. We have included the information you requested as follows:

- In section 2.4.3 “Recruitment procedure and inclusion”: We have noted the advantage of including study participants and collaborators from different geographical areas in Germany when conducting an online study, see lines 362-363.

- In section 2.5 “Data collection”: We have outlined that online data collection might impact study participants’ engagement during data collection, but is overall not inferior to in-person data collection, see lines 374-376.

- In section 7 “Discussion”: We have added people’s access to the internet as another limitation of the study, see line 717.

Page 9. You have decided to exclude SPs who are “widely known” to avoid hierarchical power dynamics in the focus groups. If those SPs are also highly experienced in the area of gender affirming SP practices, are you not worried to miss important from those? Why do you not offer individual interviews also with the SPs as with the clients? And what is the definition of “widely known” and who will decide this? There seems to be a risk of bias if this is the investigator’s decision. 

RESPONSE: Thank you for your questions. We would like to address them individually as follows:

Data collection method with the speech pathologists:

We chose focus group discussions over interviews to evoke both diversity and agreement about the research topic and to support speech pathologists to discuss the research topics. We now have rephrased our rationale to make it clearer that the focus group process can support participants to explore their perspectives on a certain topic as opposed to one-on-one settings, see lines 485-489.

Definition of the exclusion criterion “widely known” and risk of bias:

- We have changed the wording to speech pathologists who are likely to be considered “experts” by peers. We have now also specified that people are considered as experts when they are “widely known through their publications in gender affirming clinical practices”, see lines 295-298.

- To match the wording in the manuscript, we have changed the wording in the table for the inclusion and exclusion criteria to “Likely to be considered experts by peer speech pathologists”, see p. 12-13, Table 1, column “speech pathologists”, column “exclusion criteria”.

Missing information if speech pathologists “widely known” in the area are excluded: 

- Thank you for this thought. To address this, we have added that we accept this disadvantage in favour of avoiding potential conflicts of interest in the focus group discussions, see lines 299-301. 

- Further, we now have suggested that future studies could conduct one-on-one interviews with experts in the field without risking hierarchical power dynamics within peer focus groups, see lines 729-733.

Page 11, line 250. So, recruitment is already completed? Later it says that data collection is completed, confusing. 

RESPONSE: Yes, by now, recruitment and data collection have been completed. However, when we submitted early January 2024, recruitment and data collection were still in progress. We agree that this might be confusing for the reader and therefore consulted with the editor. The editor’s advice was to indicate how many study participants and collaborators are expected to be included rather than providing detail about who has already been recruited. Since this information has been added to section 2.4.2 “Sampling” (see one of you earlier comments and our reply above), this would not add any new information. Therefore, we decided to remove this section from the study protocol paper. 

Page 11, line 251 speech pathologists - SP. Earlier in the text you have written the abbreviation SP and later speech pathologist. Please be consistent. 

RESPONSE: We use “SP” as an abbreviation for “speech pathology” as a profession. We have added th

---

## [Decision Letter · Decision Letter 1]

16 Jul 2024

PONE-D-23-43876R1Gender diverse people’s psychological wellbeing and identity in the context of gender affirming speech pathology practice: A qualitative study protocolPLOS ONE

Dear Dr. Tanase,

Thank you for submitting your manuscript to PLOS ONE. After careful consideration, we feel that it has merit but does not fully meet PLOS ONE’s publication criteria as it currently stands. Therefore, we invite you to submit a revised version of the manuscript that addresses the points raised during the review process.

We look forward to receiving your revised manuscript.

Kind regards,

Marianne Clemence, Staff Editor, on behalf of,

Boshra A Arnout

Academic Editor

PLOS ONE

Journal Requirements:

Reviewers' comments:

Reviewer's Responses to Questions

**Comments to the Author**

1. Does the manuscript provide a valid rationale for the proposed study, with clearly identified and justified research questions?

Reviewer #1: Yes

Reviewer #3: Yes

Reviewer #4: Yes

2. Is the protocol technically sound and planned in a manner that will lead to a meaningful outcome and allow testing the stated hypotheses?

Reviewer #1: Yes

Reviewer #3: Yes

Reviewer #4: Yes

3. Is the methodology feasible and described in sufficient detail to allow the work to be replicable?

Reviewer #1: Yes

Reviewer #3: Yes

Reviewer #4: Yes

4. Have the authors described where all data underlying the findings will be made available when the study is complete?

Reviewer #1: Yes

Reviewer #3: Yes

Reviewer #4: Yes

5. Is the manuscript presented in an intelligible fashion and written in standard English?

Reviewer #1: Yes

Reviewer #3: Yes

Reviewer #4: Yes

6. Review Comments to the Author

You may also provide optional suggestions and comments to authors that they might find helpful in planning their study.

**Reviewer #1:** Dear authors,

thank you for your excellent work to revise the manuscript with additional information so that the readers can fully comprehend your well-planned study protocol. The additions and changes have further improved the quality of the manuscript.

I just have a few comments/questions:

Lines 354-359. All participants should have had at least 5 SP sessions for inclusion. Is this true for participants PFAB and PMAB? I do not read German and I cannot interpret the content in reference 71. Many people, also experts, do not expect that participants AFAB also need SP-training. It would be good with some clarification here.

Line 810 you name other professions beyond SP. Wouldn't it be appropriate to also mention psychotherapy?

Lines 816-817 "Furthermore gender diverse people who have had negative experiences with SP in the past might not want to participate in a study conducted by a speech pathologist." What do you base this assumtion on? Do you think this is common? Is the sentence really needed or can it be phrased differently?

I found that S6 in the appendices could be a presented as a figure as it was in the original version of the manuscript but I leave this to you to decide what you find the best.

Good luck finalizing the manuscript. I really look forward to read about the results!

**Reviewer #3:** I am very happy with the revisions. The description now clearly highlights the added benefit of the publication of the study protocol, and presents the rationale and design decisions in a clear and convincing manner.

**Reviewer #4:** A really thoughtfully written protocol on a fascinating sounding study.

All the revisions made thus far have strengthened the paper.

However, I would like to suggest the inclusion of Braun and Clarke's 2006 and 2020 papers on the practical aspects of undertaking either a reflexive thematic analysis or a codebook thematic analysis, as both are equally useful and valid but are the not the same process.

Braun, V. and Clarke, V. 2020. One size fits all? What counts as quality practice in (reflexive) thematic analysis? Qualitative Research in Psychology, 18, 328–352. https://doi.org/10.1080/14780887.2020.1769238.

Braun, V. and Clarke, V. 2006. Using thematic analysis in psychology. Qualitative Research in Psychology, 3, 77–101.

At the moment, the analysis plan reads like a mix of both so a little more clarity is needed.

Also this sentence in the manuscript is confusing: "Since RTA is not seen as suitable for the analysis of narratives [49], the data analysis approach will be expanded to enable analysing narratives in the interviews or focus group discussions". What type of 'narrative' do you mean? Reflexive thematic analysis is often used (and used well) with different interview narratives in an array of qualitative studies. This needs to be clarified.

Overall though, this is an excellent study and I look forward to the protocol being published and learning more about the findings in the future.

7. PLOS authors have the option to publish the peer review history of their article (what does this mean?). If published, this will include your full peer review and any attached files.

Reviewer #1: No

Reviewer #3: No

Reviewer #4: No

---

## [Author Response · Author response to Decision Letter 1]

5 Aug 2024

Reviewer #1: Comments to the authors.

Dear authors,

thank you for your excellent work to revise the manuscript with additional information so that the readers can fully comprehend your well-planned study protocol. The additions and changes have further improved the quality of the manuscript.

I just have a few comments/questions:

RESPONSE: Thank you for your very kind and encouraging comment and your great help with improving the first version of our manuscript. Many thanks for your further comments and questions on the current version of our manuscript. We, once again, appreciate your thorough review. You will find our reply to each of your comments below. The pages, sections and lines we refer to in our replies are those in the revised unmarked manuscript version.

Lines 354-359. All participants should have had at least 5 SP sessions for inclusion. Is this true for participants PFAB and PMAB? I do not read German and I cannot interpret the content in reference 71. Many people, also experts, do not expect that participants AFAB also need SP-training. It would be good with some clarification here.

RESPONSE: Thank you for this comment. To address this, we have made the following changes:

- SP with people PFAB: Thank you for pointing out that some important information needs to be included. We now have added the information that gender diverse people may feel a discomfort with their vocal patterns and seek professional support form a speech pathologist and have emphasised that “This can be the case for any gender diverse person, e.g., transfeminine, transmasculine, or non-binary people, even for people whose vocal patterns may have changed through gender affirming hormones.”, see lines 98-102.

- Minimum number of SP sessions: We have clarified that the literature does not specify a particular minimum number of SP sessions for gender diverse clients except for people aiming to present as female, which is a reported minimum number of five. We have stated that we assume that this number is transferable to any gender diverse person, regardless of whether their vocal patterns have changed through gender affirming hormones. We then have concluded that therefore, the inclusion criterion of having attended at least five SP sessions will apply to any gender diverse person interested in participating in the study, see lines 297-303.

Line 810 you name other professions beyond SP. Wouldn't it be appropriate to also mention psychotherapy?

RESPONSE: Thank you for this comment. We have now made the following changes:

- Since there are few studies in psychology that explore PWB and identity in gender diverse people, we have included some more information in the background section and have stated that “Research in psychology is beginning to show that creating positive narratives around gender diverse identities can have positive effects on gender diverse people’s PWB.”, see lines 140-141. 

- To avoid evoking the impression that there are no studies in psychology on PWB and identity in gender diverse people, we have changed the statement in the part you are referring to slightly. We now have stated that future studies could centre disciplines beyond SP, such as, psychology, endocrinology, or occupational therapy and that the already existing literature in psychology could be a valuable resource when planning such studies, see lines 721-725.

Lines 816-817 "Furthermore gender diverse people who have had negative experiences with SP in the past might not want to participate in a study conducted by a speech pathologist." What do you base this assumtion on? Do you think this is common? Is the sentence really needed or can it be phrased differently?

RESPONSE: We have phrased this differently and have added references to support our statement. We have explained that due to experiences of discrimination or harassment in healthcare settings, some gender diverse people might not want to participate in a study conducted by a healthcare professional, such as a speech pathologist, see lines 730-732. 

I found that S6 in the appendices could be a presented as a figure as it was in the original version of the manuscript but I leave this to you to decide what you find the best.

RESPONSE: Thank you for this comment. We have double-checked the file format for figures (.tif). Unfortunately, this format does not allow a very high resolution of the figure. Since the PDF format has a higher resolution and is allowed as a format for supporting information files, we decided to leave the file in a PDF format. 

Good luck finalizing the manuscript. I really look forward to read about the results!

RESPONSE: Thank you again for your feedback and your help to improve our manuscript. We really appreciate your time and effort.

Reviewer #3: Comments to the authors.

I am very happy with the revisions. The description now clearly highlights the added benefit of the publication of the study protocol, and presents the rationale and design decisions in a clear and convincing manner.

RESPONSE: Thank you for your very kind and encouraging comment and for your great help with improving the first version of our manuscript. We really appreciate your time and effort. 

Reviewer #4: Comments to the authors.

A really thoughtfully written protocol on a fascinating sounding study.

All the revisions made thus far have strengthened the paper.

RESPONSE: Thank you for your very kind and encouraging comment. Many thanks for your further comments on the current version of our manuscript. We appreciate your thorough review. You will find our reply to each of your comments below. The pages, sections, lines, and reference numbers we refer to in our replies are those in the revised unmarked manuscript version.

However, I would like to suggest the inclusion of Braun and Clarke's 2006 and 2020 papers on the practical aspects of undertaking either a reflexive thematic analysis or a codebook thematic analysis, as both are equally useful and valid but are the not the same process.

Braun, V. and Clarke, V. 2020. One size fits all? What counts as quality practice in (reflexive) thematic analysis? Qualitative Research in Psychology, 18, 328–352. https://doi.org/10.1080/14780887.2020.1769238.

Braun, V. and Clarke, V. 2006. Using thematic analysis in psychology. Qualitative Research in Psychology, 3, 77–101.

At the moment, the analysis plan reads like a mix of both so a little more clarity is needed.

RESPONSE: Thank you for pointing out this important literature and for noticing that clarity is needed here. To provide more clarity, we have made the following changes in the section about data analysis:

Differentiation between codebook TA and RTA:

To highlight that RTA is the analysis method we use and to avoid the impression that we use codebook TA, we made the following changes:

- Minor changes to the wording (see lines 549-563): We now have stated that 

 o the aim of RTA is to develop themes,

 o the first author creates meaning, 

 o RTA helps guiding the process of developing themes, 

 o codes are based on collected data, 

 o codes are clustered together to generate themes

- Additional information: We have explained that themes will not be created a priori and analysis will be data driven with the first author actively constructing themes. We have further explained that this process will be shaped by the first author’s reflections about the context in which data analysis takes place (as a core element of RTA) and have referred to our reflexivity statement earlier in the manuscript, see lines 550-554.

- After further discussion within the author team, we have reviewed our implementation of member reflections. With the changes we now have made, we would like to emphasise that member reflections will not be used to validate data but will be used in a way that is in accordance with RTA. Therefore, we have made the following changes:

 o We have described that when implementing member reflections, we will send study participants “the first author’s ideas about potential initial themes constructed from the interviews or focus group discussions” and will ask study participants about their reflections on these ideas. Due to this change, we have also changed the phase in which we will be using member reflections (phase 3 instead of 1 or 2). These changes can be found on p. 25, in Table 4, row “Member reflections”. 

 o In the section about research quality, we have slightly changed the wording. Where we wrote about credibility, we now state that strategies like member reflections will be used to gain more insight into the data, see lines 664-665. 

Inclusion of further references:

We have now cited the 2020 paper by Braun and Clarke (2021 in the reference version we have) you suggested. However, even though we see the value of the 2006 paper, we decided to cite more recent work by Braun and Clarke in this section instead, particularly since Braun and Clarke partially criticise their own 2006 paper. We have cited the following literature by Braun and Clarke instead: their 2021 paper as suggested (reference number 104, cited multiple times more now), their 2022 book (reference number 52, as cited previously), and their 2023 paper (reference number 103, as cited previously). This literature contains the information needed to make the changes in the manuscript as you suggested. The reference changes we have made can be found on p. 23-24, in section 2.6.1. Note that the changes made to the references are unfortunately not visible in the track changes version of the manuscript since this is not possible with the referencing software (EndNote) we use.

Also this sentence in the manuscript is confusing: "Since RTA is not seen as suitable for the analysis of narratives [49], the data analysis approach will be expanded to enable analysing narratives in the interviews or focus group discussions". What type of 'narrative' do you mean? Reflexive thematic analysis is often used (and used well) with different interview narratives in an array of qualitative studies. This needs to be clarified.

RESPONSE: Thank you for this thought-provoking comment that prompted further discussion within the author team. We have made the following changes in the manuscript:

Definition of narratives:

- In the data collection section, where we write about narratives in episodic interviews, we have clarified what we mean by “narratives”. We have stated that narratives comprise both a story (i.e., the sequence of events) and a plot (i.e., the speaker’s attitudes, perceptions and motives) related to an episode, see lines 458-460.

- For clarity reasons, we added another brief description of narratives in the data analysis section later in the manuscript, see p. 25, Table 4, row “Memos”.

RTA and the analysis of narratives: 

Instead of stating that RTA is not suitable to analyse narratives, we have clarified how and why we will use the strategies we have listed as complementary strategies for RTA. We have made the following changes:

- To highlight that an in-depth engagement with the narrative parts of the data is optional, we have clarified that, if a more in-depth engagement with participants' stories is needed, memos will be created based on questions to engage with the narrative parts (story and plot of an episode) of the data, see p. 25. Table 4, row “Memos”. 

- Since in our previous version of the manuscript, it read as if we would use memos only for the narrative parts of the data, we have now clarified that memos, member reflections and peer debriefing aim to facilitate further engagement with the data and to support the first author to construct meaningful themes. We have referred to Table 4 for more detail, see lines 571-575. 

Overall though, this is an excellent study and I look forward to the protocol being published and learning more about the findings in the future.

RESPONSE: Thank you again for your feedback and your help to improve our manuscript. We really appreciate your time and effort.

---

## [Editor Report · Decision Letter 2]

13 Sep 2024

Gender diverse people’s psychological wellbeing and identity in the context of gender affirming speech pathology practice: A qualitative study protocol

PONE-D-23-43876R2

Dear Dr. Tanase,

We’re pleased to inform you that your manuscript has been judged scientifically suitable for publication and will be formally accepted for publication once it meets all outstanding technical requirements.

Kind regards,

Boshra A Arnout

Academic Editor

PLOS ONE
---

## [Editor Report · Acceptance letter]

23 Sep 2024

PONE-D-23-43876R2 

PLOS ONE

Dear Dr. Tanase, 

I'm pleased to inform you that your manuscript has been deemed suitable for publication in PLOS ONE. Congratulations! Your manuscript is now being handed over to our production team.

Kind regards, 

on behalf of

Professor Boshra A Arnout 

Academic Editor

PLOS ONE